# Association between Prescribers’ Perceptions of the Utilization of Medication for Opioid Use Disorder and Opioid Dependence Treatability

**DOI:** 10.3390/healthcare10091733

**Published:** 2022-09-09

**Authors:** David Adzrago, Angela Di Paola, Jialing Zhu, Alejandro Betancur, J. Michael Wilkerson

**Affiliations:** 1Center for Health Promotion and Prevention Research, CDC Prevention Research Center, School of Public Health, The University of Texas Health Science Center at Houston (UTHealth), Houston, TX 77030, USA; 2AIDS Program, Section of Infectious Diseases, Department of Internal Medicine, Yale School of Medicine, New Haven, CT 06510, USA

**Keywords:** medication for opioid use disorder, SUD treatment, prescribers’ perceptions of treatment

## Abstract

Background: Medication for opioid use disorder (MOUD) has been proven to be effective, yet the perceptions or beliefs of prescribers of MOUD may have a substantial impact on their prescribing and dispensing of MOUD and their patients’ accessibility and utilization of MOUD services. We examined the associations of the perceptions of medical and pharmacy professionals regarding MOUD with sociodemographic characteristics, personal experiences with substance use disorders, and perceptions of opioid treatment. Method: Data were collected via telephone or online survey from March to August 2021, in Texas, to assess medical and pharmacy professionals’ perceptions of MOUD. Our sample included 542 participants who completed the survey. A multinomial logistic regression analysis was conducted to assess perceptions of MOUD, its use, and their correlates. Results: The participants had a mean age of 35 years (SD = 7.13) and had worked, on average, 6.90 years (SD = 5.37) in their current positions. The majority of the participants were males (50.93%) and medical professionals (82.01%). More than one third of the participants believed MOUD did not lead to abstinence or recovery (36.16%). Those who had personal experiences with a substance use disorder were more likely to believe that MOUD could be a replacement drug for previously misused substance(s) (RRR = 2.06, 95% CI = 1.19, 3.59) and that MOUD did not lead to abstinence or recovery (RRR = 2.34, 95% CI = 1.40, 3.91). However, the risk ratio values were lower for those who believed that a stigma against MOUD was a barrier for patients initiating and adhering to MOUD (MOUD is a replacement drug for previously misused substances (initiation RRR = 0.43, 95% CI = 0.19, 0.93 and adhering RRR = 0.30, 95% CI = 0.13, 0.71) or MOUD does not lead to abstinence or recovery (initiation RRR = 0.26, 95% CI = 0.13, 0.54 and adhering RRR = 0.36, 95% CI = 0.17, 0.78)). The various perceptions of the utilization of MOUD were not statistically different between medical and pharmacy professionals. Conclusion: Perceptions, experience with substance use disorder, and stigma against the utilization of MOUD influenced negative perceptions about MOUD. An innovative strategy is needed to improve medical and pharmacy professionals’ perceptions of MOUD, while efforts are being made to promote the use of MOUD for patients with opioid use disorders.

## 1. Introduction

Medication for opioid use disorder (MOUD), in addition to counseling and behavioral therapies for treating substance use disorders (SUDs), are effective for treating opioid use disorder (OUD) [1,2,3,4]. These medications include buprenorphine/naloxone, methadone, and naltrexone, which are known to be effective for managing and reducing OUD withdrawal symptoms, psychological cravings, and functioning among individuals in recovery [3,4,5]. The utilization of MOUD doubles opioid abstinence, reduces opioid use, and increases MOUD retention in recovery treatment [5]. MOUD reduces the risk of opioid overdose mortality significantly (0.24 per 100 person-years) as compared with those that are untreated (2.43 per 100 person-years) [3]. Furthermore, the risks of all-cause mortality and overdose mortality were higher among untreated and discharged patients than among patients receiving MOUD [3].

Despite the effectiveness of MOUD in reducing and preventing OUD and its associated health consequences, prescribers’ perceptions and familiarity with MOUD continue to barricade the prescription, dispensation, and dissemination of MOUD [3,6,7]. In a sample of medical professionals from two national symposia on opioid dependence, some prescribers perceived or believed that patients diverted the prescribed medications (i.e., naloxone, methadone, and naltrexone) for other uses, to get high, or shared them with friends or sold the medications, which increased misuse and accidental overdoses of these medications. The prescribers also believed that the diversion of patients’ medications discouraged professional help-seeking [7]. In addition, they believed that there was a lack of access to affordable local MOUD and lack/poor insurance coverage of MOUD services [7]. Another study of medical prescribers undergoing MOUD training reported that participants held positive attitudes or perceptions about MOUD. The participants believed that MOUD was effective, saved patient lives, improved patient health, had minimal side effects, reduced/prevented cravings and withdrawal symptoms, and was easy to administer [8]. Stigma about OUD discourages prescribers from providing effective treatments to patients, including MOUD [9].

The reluctance of prescribers to treat patients with OUD also limits access to MOUD services, especially those in rural settings [9], leading to most providers or prescribers not actively offering MOUD services [9,10,11,12]. Potential reasons for the unwillingness of providers or prescribers to offer MOUD included their lack of confidence in and ability to treat patients with SUD, including OUD, and stigma about these patients and MOUD [10,13]. Some providers or prescribers also perceived MOUD as replacing one substance with another substance and, as a result, believed the better treatment of OUD was abstinence as compared with MOUD [10,13]. Additionally, fear of possible lawsuits due to inappropriate prescriptions could also be a reason for their reluctance [14,15]. Inappropriate prescriptions have affected the physical and mental health of Americans and have led to many lawsuits against many healthcare facilities and providers [14,15].

The above review of the literature suggests that MOUD is effective for treating OUD, but the perceptions or beliefs of prescribers about MOUD may have a substantial impact on the prescription and dispensation of MOUD, as well as on patients’ accessibility and utilization of MOUD services. However, there is still limited and evolving research in the literature on prescribers’ perceptions of MOUD and OUD treatability. Thus, we conducted this study: (1) to estimate differences in the perceptions of medical and pharmacy professionals about MOUD and OUD treatability and (2) to examine the associations of perceptions of medical and pharmacy professionals about MOUD with sociodemographic characteristics, personal experiences with substance use disorder, perceptions about specific medications, and perceptions about opioid treatment.

## 2. Methods

### 2.1. Study Design and Participants

A cross-sectional survey was conducted from March to August 2021 to assess physicians’ and pharmacists’ attitudes toward and knowledge of MOUD. The participants were specifically asked about their attitudes and beliefs concerning the Naloxone Standing Order Training for Certification and Dispensing of Naloxone/Narcan, and their attitudes and beliefs about dispensing this harm-reduction medication to their patients. The data were collected through telephone and online REDCap surveys. The participants were selected at their respective pharmacies where they worked in Houston and Harris County region, Texas. The survey included only small, privately owned pharmacies in the Houston and Harris County region. The large pharmacies were not included in the study as their corporate policy prohibited them from participating in research without authorization.

The first 250 participants to complete the survey and provide an e-mail address at the end of the survey were sent 25 USD Amazon e-gift card via e-mail as compensation for participation. The expected sample size was 250, but 557 participants completed the survey. Our analytical sample included 542 participants who had complete data on the perception of the MOUD questions.

### 2.2. Measures

Our dependent variable was the prescribers’ perceptions of MOUD. The participants were asked to indicate which of the following statements they most agreed with: (1) MOUD helps patients during their recovery. (2) MOUD is just a replacement drug for patients’ previous use of illicit drugs. (3) Individuals using MOUD are not technically abstinent. (4) Individuals using MOUD are abstinent as long as these individuals are not using the drug they abused. (5) Individuals can never truly be recovered if they continue using MOUD. We categorized their responses into MOUD helps patients during their recovery (Options 1 and 4), MOUD is a replacement drug for previously misused substances (Option 2), and MOUD does not lead to abstinence or recovery (Options 3 and 5).

The explanatory variables had 5-point Likert agreement response options from strongly disagree to agree strongly and were categorized as (1) disagree (i.e., strongly disagree or somewhat disagree), (2) neutral, and (3) agree (i.e., somewhat agree or strongly agree). Perceptions about opioid dependence treatability were measured by asking the participants whether “OUD is a treatable illness.” Up-to-date on OUD treatment using buprenorphine was assessed by responses to the statement, “I am up-to-date on the literature regarding the efficacy and safety of buprenorphine prescriptions for opioid use disorders.” Access to buprenorphine reduces opioid overdoses and deaths was based on responses to the statement “Increased access to buprenorphine can help reduce opioid overdoses and deaths.” Similar items included responses about their level of agreement regarding the perception of stigma against MOUD as a barrier to patients’ initiating and adhering to MOUD. Other explanatory variables included a personal experience with substance use disorder, which was based on the question, “Do you have personal experience with substance use disorder (yourself or a loved one)?” (yes/no).

Items regarding beliefs about naloxone/narcan included whether or not (yes/no) naloxone/narcan: increases drug use, decreases drug use, does not change drug use rates, should be available in areas with high rates of opioid use, should be available everywhere, should not be available, should be available with insurance coverage, and should be available for free.

Participants’ sociodemographic characteristics included age, gender identity (male or female), current position at work (medical professional or pharmacy professional), years worked in current position, and rural-urban classification of workplace location (urban, suburban, or rural).

### 2.3. Statistical Analysis

We used Stata 16.1 to perform the statistical analysis (Stata Corp., College Station, TX, USA). Frequencies with the corresponding percentages and means (with standard deviation [SD]) were reported for the univariate statistics. Differences in the groups were determined using Pearson chi-square tests (for categorical variables) and ANOVA (for continuous variables) for the bivariate analyses with a statistical significance level of *p* < 0.05. The associations between the dependent and the explanatory variables were examined using a multinomial logistic regression analysis. Relative risk ratio (RRR) values with a 95% confidence interval (95% CI) were reported for the multinomial logistic regression analysis, and significance was determined at *p* < 0.05.

## 3. Results

The participants had a mean age of 35 years (SD = 7.13) and worked, on average, 6.90 years (SD = 5.37) in their current position. About half of the participants were female (49.07%). A majority of participants were medical professionals (82.01%), more than half of whom work in an urban area (53.69%). A third of the participants believed that MOUD did not lead to abstinence or recovery (36.16%).

The bivariate analysis results, in Table 1, showed significant differences in perception about MOUD among subgroups of the participants. A greater proportion of females than males (40.15%) believed that MOUD helped patients during their recovery. Nearly a third (37.59%) of males believed MOUD did not lead to abstinence or recovery. The perception that MOUD did not lead to abstinence or recovery was higher among those who disagreed that OUD was treatable (50.00%), had a personal experience with substance use disorder (44.29%), disagreed that access to buprenorphine reduced opioid overdoses and deaths (53.85%), disagreed that stigma against MOUD was a barrier for patients initiating MOUD (52.99%) or adhering to MOUD (48.98%), and indicated that naloxone/narcan should not be available (55.71%).

The multinomial logistic regression analysis results are summarized in Table 2. The reference group for the dependent variable, i.e., prescribers’ perceptions of MOUD, was believing that MOUD help patients during their recovery. As compared with males, females had a lower risk of believing that MOUD could be a replacement drug for previous use of illicit substances (RRR = 0.54, 95% CI = 0.31, 0.92). The number of years worked in a current position was significantly associated with lower risks of believing that MOUD could be a replacement drug for previously misused substance(s) (RRR = 0.90, 95% CI = 0.84, 0.96) as compared with believing that MOUD helped patients during their recovery. Those who were neutral about OUD being treatable, as compared with those who disagreed, had higher risks of believing that MOUD could be a replacement drug for previously misused substance(s) (RRR = 2.51, 95% CI = 1.03, 6.14).

Participants who had personal experiences with substance use disorder (versus no experience) had higher risk ratio values for believing that MOUD could be a replacement drug for previously misused substances (RRR = 2.06, 95% CI = 1.19, 3.59) or that MOUD did not lead to abstinence or recovery (RRR = 2.34, 95% CI = 1.40, 3.91). Higher risk ratio values were also observed for those who indicated that naloxone/narcan should not be available (versus should be available); MOUD could be a replacement drug for previously misused substance(s) (RRR = 4.07, 95% CI = 1.84, 9.02), or MOUD did not lead to abstinence or recovery (RRR = 5.79, 95% CI = 2.80, 12.00).

The risks were, however, lower for those who agreed that stigma against MOUD was a barrier for patients initiating MOUD (versus disagreed); MOUD could be a replacement drug for previously misused substance(s) (RRR = 0.43, 95% CI = 0.19, 0.93); or MOUD did not lead to abstinence or recovery (RRR = 0.26, 95% CI = 0.13, 0.54). Being neutral about stigma against MOUD being a barrier for patients adhering to MOUD (versus disagree) was associated with lower risks of believing that MOUD could be a replacement drug for previously misused substance(s) (RRR = 0.30, 95% CI = 0.13, 0.71) or MOUD did not lead to abstinence or recovery (RRR = 0.36, 95% CI = 0.17, 0.78). The participants who reported that naloxone/narcan should be available in areas with high rates of opioid use (versus should not be available) had a lower risk of believing that MOUD did not lead to abstinence or recovery (RRR = 0.54, 95% CI = 0.30, 0.99).

## 4. Discussion

We estimated the extent to which the demographic differences of medical and pharmacy professionals were associated with differences in perceptions about MOUD. More than one third of the participants believed that MOUD did not lead to abstinence or recovery, especially males, pharmacy professionals, and those who had experiences with substance use disorder. If a third or more of medical and pharmacy professionals do not believe MOUD is useful to persons in recovery, and they might be less likely to endorse its use when interacting with patients who have an OUD. Provider-targeted interventions that change perception are needed.

Similar to previous studies [7,16], we observed a gender identity disparity in the perception of MOUD. As compared with their male counterparts, female medical or pharmacy professionals were about 46% less likely to believe that MOUD could be a replacement drug for previously misused substances. This disparity may need further examination to understand potential factors contributing to it.

The number of years worked in a current position as a medical or pharmacy professional was negatively associated with the perceptions of utilization of MOUD. As the years worked in a current position increased, there was a lower likelihood of believing MOUD could be a replacement drug for previously misused substances. These differences in perception about MOUD based on years worked in a current role may be attributed to years of experience, and practice and perception about OUD.

Our findings further revealed that medical and pharmacy professionals who were uncertain (versus certain) that OUD was treatable were about three times more likely to believe that MOUD could be a replacement drug for previously misused substances. [16], for example, found analogous results that suggested that prescribers with more years of practice experience were less likely to perceive MOUD as a replacement substance for use with another dependence. Notably, prescribers with more years of practice are more likely to have a positive perception of OUD and its treatment, including MOUD [7,16,17]. Consequently, these prescribers would be less likely to stigmatize patients with OUD and more likely to initiate MOUD for these patients with OUD. Increased experiences, awareness, and education/training of prescribers about OUD and the use of MOUD may help to reduce and to prevent stigma about MOUD among the prescribers, thereby, improving OUD patient care and access to utilization of MOUD seervices [18,19,20,21].

Prescribers’ personal experiences with a substance use disorder, including OUD, were a significant determinant of their perceptions of MOUD. Those who had personal experiences with a substance use disorder had negative perceptions of utilization of MOUD. They were more likely to believe that MOUD could be a replacement drug for previously misused substances or that MOUD did not lead to abstinence or recovery. While there is extensive research demonstrating MOUD is effective and efficacious for OUD treatment, the negative perceptions of prescribers about the utilization of MOUD could discourage them from prescribing MOUD to their patients with OUD seeking treatment [16,22,23]. The negative perceptions among the prescribers could be due to their lack of awareness or education about the effectiveness of MOUD [24]. These negative perceptions could also discourage patients from accessing MOUD [22]. The prescribers’ negative perceptions might be emanating from their stigma against the utilization of MOUD services. Comparable with other studies [22,23], our findings revealed that prescribers who believed or were uncertain that stigma against MOUD was a barrier for patients initiating or adhering to MOUD were less likely to either believe that MOUD could be a replacement drug for previously misused substances or MOUD did not lead to abstinence or recovery. These findings imply that prescribers who have less or no stigma against patients using MOUD would also have less or no negative perception about utilization of MOUD services, suggesting the need to improve the stigma about MOUD among medical and pharmacy professionals.

Perceptions about the utilization of MOUD or the stigma against MOUD could be related to specific medication [22,23]. Whereas the perception of buprenorphine did not significantly explain the prescribers’ perceptions of the utilization of MOUD, beliefs regarding naloxone/narcan negatively and significantly influenced their perceptions of the utilization of MOUD. Those who believed that naloxone/narcan should not be available (versus should be available) were about four to six times more likely to believe that MOUD could be a replacement drug for previously misused substances or that MOUD did not lead to abstinence or recovery. Similarly, those who indicated that naloxone should be available in areas with high rates of opioid use (versus should not be available) were about 46% less likely to believe that MOUD did not lead to abstinence or recovery. This stigma can serve as a barrier to the delivery of and patient access to MOUD such as naloxone/narcan [9,16,22], although the effectiveness and efficacy of naloxone/narcan in treating opioid overdose and preventing death have been extensively documented in the literature [3,5,22,23,25]. Potential strategies to address these stigmas and perceptions may include targeted OUD and MOUD education early in medical and pharmacy training to increase knowledge and awareness about the need to treat and reduce or to prevent OUD. Specific strategies may highlight MOUDs’ evidence-based effectiveness and efficacy and emphasize that OUD is a treatable condition that deserves attention similar to other chronic conditions [26,27].

Despite the strengths of our study in providing information on prescribers’ perceptions about MOUD, there are some limitations that need to be considered. Our study participants were only from smaller clinics and privately owned pharmacies, potentially limiting the generalizability of our findings to the general medical and pharmacy professionals. The large clinical and pharmaceutical businesses were hesitant to provide the research team access to their employees. In addition, because this study was cross-sectional, we could not establish causal relationships. To expand upon the findings from this analysis, future researchers should examine the reasons behind participants’ perceptions about MOUD to identify the most modifiable factors and inform intervention development.

## 5. Conclusions

The negative perceptions about MOUD among medical and pharmacy professionals included lower years of experience, being uncertain about opioid treatability, personal experience with substance use disorders, negative perceptions about naloxone/narcan, and stigma against the effectiveness of using MOUD among OUD patients. An innovative strategy is needed to improve medical and pharmacy professionals’ perceptions of and stigma against the utilization of MOUD services, while efforts are being made to promote using MOUD for patients with OUDs. Opportunities exist for jurisdictions to leverage existing resources to change perceptions. Examples of resources that could be leveraged include developing and implementing targeted print and digital media campaigns, offering continuing medical education classes, and training community health workers and peer navigators to engage healthcare providers in conversations about MOUD. Additional research is needed to understand how best to develop an intervention to increase positive perceptions of MOUD.

## Figures and Tables

**Table 1 healthcare-10-01733-t001:** Descriptive and bivariate analyses of perceptions of people in the medical and pharmacy fields about medication for opioid use disorder (MOUD) and its correlates.

	Overall	MOUD Helps Patients during Their Recovery	MOUD Is a Replacement Drug for Previous Use of Illicit Drugs	MOUD Does Not Lead to Abstinence or Recovery	
	N (%)	*n* (%)	*n* (%)	*n* (%)	*p*-Value
**Overall**	542 (100%)	191 (35.24)	155 (28.60)	196 (36.16)	
**Age (Mean, SD)**	530 (34.66; 7.13)	188 (36.23; 9.05)	150 (32.83; 5.72)	192 (34.55; 5.50)	**<0.001**
**Gender identity**					**0.048**
Male	274 (50.93)	84 (30.66)	87 (31.75)	103 (37.59)	
Female	264 (49.07)	106 (40.15)	65 (24.62)	93 (35.23)	
**Current position at work**					0.150
Medical professional	433 (82.01)	153 (35.33)	127 (29.33)	153 (35.33)	
Pharmacy professional	95 (17.99)	35 (36.84)	19 (20.00)	41 (43.16)	
**Years worked in current position (Mean; SD)**	529 (6.90; 5.37)	188 (7.50; 6.57)	149 (5.57; 4.28)	192 (7.35; 4.63)	**0.002**
**Rural-urban classification of workplace location**					0.090
Urban	291 (53.69)	113 (38.83)	71 (24.40)	107 (36.77)	
Suburban	203 (37.45)	61 (30.05)	66 (32.51)	76 (37.44)	
Rural	48 (8.86)	17 (35.42)	18 (37.50)	13 (27.08)	
**Perception about OUD treatability**					**0.009**
Disagree	84 (15.50)	26 (30.95)	16 (19.05)	42 (50.00)	
Neutral	145 (26.75)	42 (28.97)	49 (33.79)	54 (37.24)	
Agree	313 (57.75)	123 (39.30)	90 (28.75)	100 (31.95)	
**Personal experience with SUD**					**<0.001**
No	241 (45.47)	116 (48.13)	61 (25.31)	64 (26.56)	
Yes	289 (54.53)	71 (24.57)	90 (31.14)	128 (44.29)	
**Up-to-date on OUD treatment using buprenorphine**					**0.028**
Disagree	109 (20.30)	44 (40.37)	25 (22.94)	40 (36.70)	
Neutral	164 (30.54)	48 (29.27)	43 (26.22)	73 (44.51)	
Agree	264 (49.16)	98 (37.12)	85 (32.20)	81 (30.68)	
**Access to buprenorphine reduces opioid overdoses and deaths**					**0.006**
Disagree	52 (9.63)	12 (23.08)	12 (23.08)	28 (53.85)	
Neutral	131 (24.26)	37 (28.24)	47 (35.88)	47 (5.88)	
Agree	357 (66.11)	141 (39.50)	96 (26.89)	120 (33.61)	
**Naloxone increases drug use**					0.506
No	504 (92.99)	179 (35.52)	141 (27.98)	184 (36.51)	
Yes	38 (7.01)	12 (31.58)	14 (36.84)	12 (31.58)	
**Naloxone decreases drug use**					0.855
No	345 (63.65)	122 (35.36)	96 (27.83)	127 (36.81)	
Yes	197 (36.35)	69 (35.03)	59 (29.95)	69 (35.03)	
**Naloxone does not change drug use rates**					0.174
No	341 (62.92)	120 (35.19)	106 (31.09)	115 (33.72)	
Yes	201 (37.08)	71 (35.32)	49 (24.38)	81 (40.30)	
**Naloxone should be available in areas with high rates of opioid use**					**<0.001**
No	375 (69.19)	112 (29.87)	109 (29.07)	154 (41.07)	
Yes	167 (30.81)	79 (47.31)	46 (27.54)	42 (25.15)	
**Naloxone should be available everywhere**					0.139
No	295 (54.43)	93 (31.53)	90 (30.51)	112 (37.97)	
Yes	247 (45.57)	98 (39.68)	65 (26.32)	84 (34.01)	
**Naloxone should not be available**					**<0.001**
No	402 (74.17)	173 (43.03)	111 (27.61)	118 (29.35)	
Yes	140 (25.83)	18 (12.86)	44 (31.43)	78 (55.71)	
**Naloxone should be available with insurance coverage**					0.185
No	359 (66.24)	128 (35.65)	110 (30.64)	121 (33.70)	
Yes	183 (33.76)	63 (34.43)	45 (24.59)	75 (40.98)	
**Naloxone should be available for free**					**0.001**
No	486 (89.67)	159 (32.72)	145 (29.84)	182 (37.45)	
Yes	56 (10.33)	32 (57.14)	10 (17.86)	14 (25.00)	
**Stigma against MOUD is a barrier for patients initiating MOUD**					**<0.001**
Disagree	117 (22.20)	24 (20.51)	31 (26.50)	62 (52.99)	
Neutral	149 (28.27)	42 (28.19)	43 (28.86)	64 (42.95)	
Agree	261 (49.53)	119 (45.59)	75 (28.74)	67 (25.67)	
**Stigma against MOUD is a barrier for patients adhering to MOUD**					**0.006**
Disagree	98 (18.53)	23 (23.47)	27 (27.55)	48 (48.98)	
Neutral	152 (28.73)	58 (38.16)	35 (23.03)	59 (38.82)	
Agree	279 (52.74)	107 (38.35)	87 (31.18)	85 (30.47)	

**Table 2 healthcare-10-01733-t002:** Multinomial logistic regression analysis of the perceptions of people in the medical and pharmacy fields about medications for opioid use disorder (MOUD) and its correlates.

	MOUD Is a Replacement Drug for Previous Use of Illicit Drugs		MOUD Does Not Lead to Abstinence or Recovery	
	Base/Reference Outcome: MOUD Helps Patients during Their Recovery			
	RRR	95% CI	RRR	95% CI
**Gender identity**				
Male	Ref			
Female	0.54 *	(0.31, 0.92)	0.75	(0.46, 1.25)
**Current position at work**				
Medical professional	Ref			
Pharmacy professional	0.48	(0.22, 1.03)	0.91	(0.48, 1.71)
**Years worked in current position**	0.90 **	(0.84, 0.96)	0.96	(0.92, 1.01)
**Rural-urban classification of workplace location**				
Urban	Ref			
Suburban	1.41	(0.78, 2.53)	1.01	(0.59, 1.74)
Rural	1.45	(0.56, 3.75)	0.62	(0.23, 1.63)
**Perception about opioid dependence treatability**				
Disagree	Ref			
Neutral	2.51 *	(1.03, 6.14)	0.90	(0.41, 1.97)
Agree	1.76	(0.75, 4.11)	0.85	(0.41, 1.75)
**Personal experience with substance use disorder**				
No	Ref			
Yes	2.06 **	(1.19, 3.59)	2.34 **	(1.40, 3.91)
**Up-to-date on OUD treatment using buprenorphine**				
Disagree	Ref			
Neutral	1.31	(0.57, 3.01)	2.07	(0.97, 4.42)
Agree	1.83	(0.85, 3.96)	1.64	(0.79, 3.41)
**Access to buprenorphine reduces opioid overdoses and deaths**				
Disagree	Ref			
Neutral	1.50	(0.47, 4.73)	0.69	(0.24, 1.94)
Agree	0.90	(0.31, 2.62)	0.70	(0.27, 1.80)
**Naloxone increases drug use**				
No	Ref			
Yes	1.14	(0.40, 3.23)	0.78	(0.28, 1.20)
**Naloxone decreases drug use**				
No	Ref			
Yes	1.06	(0.58, 1.95)	1.00	(0.57, 1.76)
**Naloxone does not change drug use rates**				
No	Ref			
Yes	0.89	(0.48, 1.64)	0.87	(0.49, 1.52)
**Naloxone should be available in areas with high rates of opioid use**				
No	Ref			
Yes	0.54	(0.28, 1.03)	0.54 *	(0.30, 0.99)
**Naloxone should be available everywhere**				
No	Ref			
Yes	0.69	(0.37, 1.29)	0.98	(0.55, 1.73)
**Naloxone should not be available**				
No	Ref			
Yes	4.07 **	(1.84, 9.02)	5.79 ***	(2.80, 12.00)
**Naloxone should be available with insurance coverage**				
No	Ref			
Yes	0.88	(0.48, 1.62)	1.29	(0.74, 2.24)
**Naloxone should be available for free**				
No	Ref			
Yes	0.53	(0.19, 1.44)	0.96	(0.42, 2.22)
**Stigma against MOUD is a barrier for patients initiating MOUD**				
Disagree	Ref			
Neutral	0.89	(0.39, 2.01)	0.70	(0.33, 1.46)
Agree	0.43 *	(0.19, 0.93)	0.26 ***	(0.13, 0.54)
**Stigma against MOUD is a barrier for patients adhering to MOUD**				
Disagree	Ref			
Neutral	0.30 **	(0.13, 0.71)	0.36 **	(0.17, 0.78)
Agree	0.75	(0.34, 1.65)	0.62	(0.30, 1.28)

RRR, Relative risk ratio; Ref., Reference group; 95% CI, 95% confidence interval. Statistically significant at * *p* < 0.05, ** *p* < 0.01, and *** *p* < 0.001.

## Data Availability

Our data and methods used in this research are presented in the paper. Any additional request for the data should be directed to J. Michael Wilkerson.

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
