# Peer review of "Association between Prescribers’ Perceptions of the Utilization of Medication for Opioid Use Disorder and Opioid Dependence Treatability"

_healthcare, 2022, doi:10.3390/healthcare10091733_

Round 1

Reviewer 1 Report

Healthcare -1878587– Comments for the Authors

Association between prescribers’ perception of medication for opioid use disorder use and opioid dependence treatability

General Comment: This is an important and timely topic. At the same time, each patient is unique and will benefit from different levels of care such as outpatient counseling, intensive outpatient treatment, inpatient treatment, or long-term therapeutic communities. Some patients treated in these settings may need access to OUD medications. This is why precision/personalized medicine is essential. See the below comments. 

Introduction

General Comment: The authors provided reasons e.g., stigma for prescribers “reluctance to treat patients with OUD.” Studies have shown that inappropriate prescribing is affecting the physical and mental health of Americans. This ​has led to lawsuits against many healthcare facilities and providers. Could this also be a contributing factor?

Page 2, Line 63: remove “the participants held positive attitudes or perceptions about MOUD.”

Comment: remove “the”

Methods

Page 3, Line 99-102: “The first 250 participants to complete the survey and provide an e-mail address at the end of the survey were sent a $25 Amazon e-gift card via e-mail as compensation for participation. The expected sample size was 250, but 557 participants completed the survey.”

Comment: ​Why did you compensate only the first 250 participants? I am glad the authors far exceed their sample size, but I worry about the ethics of selection. How did you end up exceeding your target by over 75%? Please clarify.

Discussion

​The authors mentioned several strategies to address providers' ​stigmas and perceptions such as ​OUD and MOUD education early in medical and pharmacy training to increase knowledge and awareness about the need to treat and reduce or prevent OUD. Even though the authors did not access this, I think it is essential for them to acknowledge its role and impact and recommend future studies to examine it.  

Author Response

Reviewer 1

General Comment: This is an important and timely topic. At the same time, each patient is unique and will benefit from different levels of care such as outpatient counseling, intensive outpatient treatment, inpatient treatment, or long-term therapeutic communities. Some patients treated in these settings may need access to OUD medications. This is why precision/personalized medicine is essential. See the below comments. 

Response

Thank you for the compliments!

Comment

Introduction

General Comment: The authors provided reasons e.g., stigma for prescribers “reluctance to treat patients with OUD.” Studies have shown that inappropriate prescribing is affecting the physical and mental health of Americans. This ​has led to lawsuits against many healthcare facilities and providers. Could this also be a contributing factor?

Response

We appreciate this recommendation. We agree that this could be a contributing factor to their reluctance. We have included this information as recommended.

Comment

 Page 2, Line 63: remove “the participants held positive attitudes or perceptions about MOUD.”

Comment: remove “the”

Response

Thank you for pointing out this error. We have removed the error.

Comment

 Methods

Page 3, Line 99-102: “The first 250 participants to complete the survey and provide an e-mail address at the end of the survey were sent a $25 Amazon e-gift card via e-mail as compensation for participation. The expected sample size was 250, but 557 participants completed the survey.”

Comment: ​Why did you compensate only the first 250 participants? I am glad the authors far exceed their sample size, but I worry about the ethics of selection. How did you end up exceeding your target by over 75%? Please clarify.

Response

We appreciate your comments. The expected sample size for the study was 250, and budgeted thusly. It was noted in the consent process that only the first 250 would be compensated, yet we had an overwhelming response of 557. Given the importance of the topic of this project, we thought it was important to include more than the anticipated 250 participants.

Comment

Discussion

​The authors mentioned several strategies to address providers' ​stigmas and perceptions such as ​OUD and MOUD education early in medical and pharmacy training to increase knowledge and awareness about the need to treat and reduce or prevent OUD. Even though the authors did not access this, I think it is essential for them to acknowledge its role and impact and recommend future studies to examine it.  

Response

Thank you for the comment. We agree that exploring early education is important, but given the extensive literature on OUD and the effectiveness of MOUD for treating OUD, this was beyond the scope of our project. We will keep this in mind for future studies.

Reviewer 2 Report

Authors have tried to find out the association between prescribers’ perception of medication for 2 Opioid Use Disorder Use and Opioid Dependence Treatability. This is an interesting topic. The study design is well and study was conducted in a proper way. My specific comments are mentioned below:

1. How sample size was calculated?

2. The questionnaire used for the study should be validated.

3. References should be cross-checked.

4. Manuscript should be revised carefully for grammatical and typographical errors. 

Author Response

Reviewer 2

Authors have tried to find out the association between prescribers’ perception of medication for 2 Opioid Use Disorder Use and Opioid Dependence Treatability. This is an interesting topic. The study design is well and study was conducted in a proper way. My specific comments are mentioned below:

Response

We appreciate your compliments!

Comment

  1. How sample size was calculated?

Given this study was an exploratory study of beliefs, traditional sample size calculations based on effect size was not conducted. However, researchers wanted to ensure saturation of beliefs by having a robust sample of 250 participants.

Comment

  1. The questionnaire used for the study should be validated.

Response

Thank you for your comment, items for this study were not validated given the unique beliefs explored.

Comment

  1. References should be cross-checked.

Response

Thank you for this comment. We have cross-checked our references.

Comment

  1. Manuscript should be revised carefully for grammatical and typographical errors. 

Response

Thank you for this comment. We have carefully reviewed and revised the manuscript for grammatical and typographical errors.